# 2,4-Dichlorophenoxyacetic Acid Induces Degeneration of mDA Neurons In Vitro

**DOI:** 10.3390/biomedicines11112882

**Published:** 2023-10-24

**Authors:** Tamara Russ, Lennart Enders, Julia M. Zbiegly, Phani Sankar Potru, Johannes Wurm, Björn Spittau

**Affiliations:** 1Medical School OWL, Anatomy and Cell Biology, Bielefeld University, 33615 Bielefeld, Germany; tamara.russ@uni-bielefeld.de (T.R.);; 2Institute of Anatomy, University of Rostock, 18051 Rostock, Germany; 3Institute for Anatomy and Cell Biology, Department of Molecular Embryology, Faculty of Medicine, University of Freiburg, 79106 Freiburg, Germanyjmz28@cam.ac.uk (J.M.Z.); 4CeMM Research Center for Molecular Medicine of the Austrian Academy of Sciences, 1090 Vienna, Austria; 5UK Dementia Research Institute and Department of Clinical Neurosciences, University of Cambridge, Cambridge Biomedical Campus, Cambridge CB2 0SL, UK

**Keywords:** 2,4-Dichlorophenoxyacetic acid, microglia, mDA neurons, neurodegeneration, TNFα

## Abstract

**Background:** Parkinson’s disease (PD) affects 1–2% of the population over the age of 60 and the majority of PD cases are sporadic, without any family history of the disease. Neuroinflammation driven by microglia has been shown to promote the progression of midbrain dopaminergic (mDA) neuron loss through the release of neurotoxic factors. Interestingly, the risk of developing PD is significantly higher in distinct occupations, such as farming and agriculture, and is linked to the use of pesticides and herbicides. **Methods:** The neurotoxic features of 2,4-Dichlorophenoxyacetic acid (2,4D) at concentrations of 10 µM and 1 mM were analyzed in two distinct E14 midbrain neuron culture systems and in primary microglia. **Results:** The application of 1 mM 2,4D resulted in mDA neuron loss in neuron-enriched cultures. Notably, 2,4D-induced neurotoxicity significantly increased in the presence of microglia in neuron-glia cultures, suggesting that microglia-mediated neurotoxicity could be one mechanism for progressive neuron loss in this in vitro setup. However, 2,4D alone was unable to trigger microglia reactivity. **Conclusions:** Taken together, we demonstrate that 2,4D is neurotoxic for mDA neurons and that the presence of glia cells enhances 2,4D-induced neuron death. These data support the role of 2,4D as a risk factor for the development and progression of PD and further suggest the involvement of microglia during 2,4D-induced mDA neuron loss.

## 1. Introduction

The degeneration of midbrain dopaminergic (mDA) neurons and the subsequent decrease in striatal dopamine (DA) levels are the hallmarks of Parkinson´s disease (PD) that lead to typical clinical symptoms such as akinesia/bradykinesia, resting tremor, rigidity and gait disturbances [1]. PD affects 1–2% of people over 60 years of age [2,3] and the majority of PD cases are sporadic, without any family history of the disease. As the underlying reasons for the loss of midbrain DA neurons are unknown, these cases are thus referred to as idiopathic [4]. A common histopathological feature of virtually all PD cases is a neuroinflammatory response driven by microglia, which are the resident innate immune cells of the central nervous system (CNS) [5]. Genetic, as well as toxin-based, animal models for PD have further confirmed that increased microglia reactivity triggers the onset and progression of midbrain DA neuron loss through the release of neurotoxic factors such as TNFα [6,7]. Interestingly, the risk of developing PD is significantly higher in distinct occupations, such as farming and agriculture, and is linked to the use of pesticides and herbicides [8]. 2,4-Dichlorophenoxyacetic acid (2,4D) and its derivates are worldwide-used herbicides used to limit the growth of broad leaf weeds. However, exposure to 2,4D has been reported to induce an adverse range of health side effects, including embryotoxicity, teratogenic effects, and neurotoxicity [9,10,11]. Further in vitro studies have underlined the neurotoxic and apoptotic effects of 2,4D in cerebellar granule cells [12,13] and described disruption of the cytoskeleton and disorganization of the Golgi apparatus as the underlying molecular mechanisms of 2,4D-mediated neurotoxicity [14]. Intracerebral injections of 2,4D into rat brains resulted in region-specific neurotoxicity in the nigrostriatal system [15]. Together, these data suggest that dopaminergic neurons might be more susceptible to 2,4D-induced neurotoxic effects. Although intracerebral injections of 2,4D induced locomotor impairments, behavioral alterations and developmental disturbances of the monoamine neurotransmitter system [15,16,17], the degeneration of mDA neurons has not been reported [15]. 

In the present preliminary study, we therefore analyzed the neurotoxic features of 2,4D in two distinct E14 midbrain neuron culture systems. Whereas neuron-enriched cultures were used to detect the direct 2,4D neurotoxic effects, neuron-glia cultures were employed to understand the contribution of microglia reactivity triggered by 2,4D-induced neurodegeneration in vitro. Our data clearly demonstrate that mDA neuron loss after 2,4D application is significantly increased in the presence of microglia in neuron-glia cultures, indicating microglia-mediated neurotoxicity as a key mechanism for progressive neuron loss in this in vitro setup. However, 2,4D alone was not capable of directly triggering primary microglia reactivity. Taken together, we provide evidence that 2,4D is neurotoxic for mDA neurons and that the presence of microglia might enhance 2,4D-induced neuron death, thereby underlining the importance of 2,4D as a risk factor for the development and progression of PD and in further strengthening the impact of microglia-mediated neurotoxicity on mDA neuron loss.

## 2. Materials and Methods

### 2.1. Mice

Time pregnant (E14) NMRI mice (Janvier, Le Genest-Saint-Isle, France) were used for the generation of mDA neuron cultures. P0/P1 pups from NMRI mice were used to generate primary microglia cultures. The animal study protocol was approved by the Federal Ministry for Nature, Environment and Consumer´s Protection of the federal state of Baden-Württemberg (X12/01D and X12/02D) and by the University of Rostock and the Landesamt für Landwirtschaft, Lebensmittelsicherheit und Fischerei Mecklenburg-Vorpommern.

### 2.2. E14 Ventral Midbrain Dopaminergic Neuron-Enriched Cultures

Neuron-enriched cultures from E14 ventral midbrains were prepared as described by Spittau et al. [18]. Ventral midbrains from embryonic day 14 (E14) NMRI mice were collected and washed in ice-cold Hank´s BSS (PAA, Cölbe, Germany). Tissues were enzymatically digested using Trypsin-EDTA (PAA) for 15 min at 37 °C after the removal of meninges and blood vessels. An equal volume of ice-cold FCS and DNase (Roche, Mannheim, Germany) at a final concentration of 0.5 mg/mL was added before gentle dissociation using wide- and narrow-bored fire-polished Pasteur pipettes. Suspensions were pelleted at 1000× *g* for 3 min, and finally resuspended in culture medium. Cells from 2 midbrains were seeded on poly-D-lysine-coated glass coverslips (12 mm diameter) in 24-well plates corresponding to a density of 200.000 cells/cm^2^. Neuron-enriched cultures were maintained in serum-free DMEM/F12 (PAA, Cölbe, Germany) medium containing N2 supplements (Invitrogen, Darmstadt, Germany) and 1% Penicillin/Streptomycin (PAA) in a 5% CO_2_/95% humidified air atmosphere at 37 °C. Treatments were started on day in vitro (DIV) 1 for 48 h. Cells were finally fixed in 4% paraformaldehyde (PFA) for 20 min at room temperature (RT) and used for immunocytochemistry (ICC). 

### 2.3. E14 Ventral Midbrain Dopaminergic Neuron-Glia Cultures

Preparations of neuron-glia cultures from postnatal mouse brains have recently been described [19]. Midbrains were dissected as described for neuron-enriched E14 ventral midbrain cultures. Resuspension of dissociated cells was performed using DMEM/F12 medium containing 10% FCS, 10% Horse serum (HS) and 1% Penicillin/Streptomycin (PAA, Cölbe, Germany). Cells were seeded on poly-D-lysine-coated glass coverslips (12 mm diameter) in 24-well plates at a density of 1.5 midbrains/coverslip corresponding to a density of 200.000 cells/cm^2^. On DIV 1, the medium was changed and the cultures were maintained in a 5% CO_2_/95% humidified air atmosphere at 37 °C for additional 7 days in order to obtain mature neuron-glia cultures. Treatments were started at DIV 8 for 48 h under serum-free conditions. Afterwards, cells were fixed in 4% paraformaldehyde (PFA) for 20 min at room temperature (RT) and used for subsequent immunocytochemistry (ICC).

### 2.4. Primary Microglia Culture

Preparations of primary microglia cultures from postnatal mouse brains have been described elsewhere [20]. Briefly, P0/P1 pups from NMRI mice were used and the brains were isolated and washed in ice-cold Hank´s BSS (PAA, Cölbe, Germany). After the removal of blood vessels and meninges, the brains were enzymatically digested using Trypsin-EDTA (PAA, Cölbe, Germany) for 15 min at 37 °C. Ice-cold FCS and DNase (Roche, Mannheim, Germany) at a final concentration of 0.5 mg/mL were added before cell dissociation using wide- and narrow-bored fire-polished Pasteur pipettes. After centrifugation, cells were collected and resuspended in DMEM/F12 medium containing 10% FCS and 1% Penicillin/Streptomycin (PAA). Cultures were maintained in poly-D-lysine-coated flasks (2–3 brains/flask) in a 5% CO_2_/95% humidified air atmosphere at 37 °C. Cultures were washed with PBS at day in vitro (DIV) 2 and 3, and fresh medium was added. Microglia were harvested after 10–14 days in culture using the shake-off technique at 250–300 rpm for 30 min. FITC-coupled isolectin (Sigma-Aldrich, Schnelldorf, Germany) was used to label the microglia and validate the microglia purity (>95%). Microglia were seeded on glass coverslips in 24-well plates and treatments were performed under serum-free conditions. All experiments in the current study involving the generation of primary cell cultures were approved by the Federal Ministry for Nature, Environment and Consumer´s Protection of the federal state of Baden-Württemberg and were conducted in accordance with the respective national, federal and institutional regulations. NMRI mice used for E14 midbrain cultures and primary microglia cultures were obtained from Janvier.

### 2.5. Cytokines and Reagents

Neuron-cultures and primary microglia were treated with 2,4-Dichlorophenoxyacetic acid (2,4D, D 7299, Sigma-Aldrich, Schnelldorf, Germany). First, 100 mg 2,4D was solved in 100 mL 95% ethanol (Carl Roth, Karlsruhe, Germany) according to the manufacturer´s instructions. Further dilutions were performed to yield a final stock concentration of 100 mM. Treatment concentrations of 10 µM and 1 mM were achieved through dilutions with cell culture medium (1:10.000 and 1:100). When using this 2,4D treatments, the ethanol concentrations were 0.043% and 0.00043%, respectively. In order to exclude direct ethanol effects, a concentration of 0.043% ethanol was used as a control (EtOH) corresponding to the 2,4D concentration of 1 mM (dilution of 1:100). Microglia cultures were treated with Interferon-γ (Peprotech, Hamburg, Germany) using concentrations of 1 ng/mL and 10 ng/mL. 

### 2.6. Immunocytochemistry

The number of tyrosine hydroxylase (TH) positive cells in E14 neuron-enriched as well as neuron-glia cultures was counted after immunocytochemistry. After fixation of cells with 4% PFA, coverslips were washed three times (5 min each) with PBS, and blocking was performed for 1 h at RT using PBS + 10% normal goat serum (Invitrogen, Darmstadt, Germany) and 0.1% Triton-X-100 (Roche). Coverslips were incubated overnight with anti-TH (1:1000, Millipore, Schwalbach, Germany) and washed three times with PBS, and goat-anti rabbit peroxidase-conjugated secondary antibodies (1:500, Dianova, Hamburg, Germany) for 1 h at RT were used. Immunoreactivity was visualized using DAB (3,3´-diaminobenzidine (Sigma-Aldrich, Schnelldorf, Germany), as described by Adams [21]. Microglia were stained using FITC-coupled Isolectin IB4 (Thermo Fisher Scientific, Darmstadt, Germany). Nuclei were counterstained using 4′6-diamidino-2-phenylindole (Dapi, Roche) and coverslips were mounted on glass slides with Fluoromount-G (Thermo Fisher Scientific). Images were captured using A Zeiss AxioImager I (Zeiss, Göttingen, Germany) and a BZ-X800 fluorescence microscope (Keyence, Neu-Isenburg, Germany). Quantifications of TH^+^ neurons were performed using phase-contrast images of TH-DAB staining. All TH^+^ cells were counted on each coverslip using the StereoInvestigator Software Version 2017.03 (MicroBrightField, Magdeburg, Germany).

### 2.7. MTT Assay

All cell viability measurements for primary microglia cultures were obtained using 3-(4,5-dimethylthiazol-2-yl)-2,5-di-phenyltetrazolium bromide (MTT, Sigma-Aldrich, Schnelldorf, Germany), as described by Mosman [22]. Microglia were plated in 24-well plates and treated accordingly. Then, 5 µL MTT dissolved in PBS (5 mg/mL) was added per 100 µL cell culture supernatant followed by incubation for 2 h at 37 °C. Afterwards, media were discarded and 100 µL isopropanol with 0.04 N HCl was added. Cell culture plates were shaken for 30 min and absorptions were measured at 570 nm with a Multiskan FC plate reader (Thermo Fischer, Darmstadt, Germany). Measurements were performed in triplicates and the results are given as percentages of control groups. 

### 2.8. TNFα and IL6 ELISA

The levels of TNFα and IL6 were quantified in supernatants from primary microglia and E14 ventral midbrain dopaminergic neuron-glia cultures using ELISA Development Kits (Peprotech, Hamburg, Germany) according to the manufacturer´s instructions. Color reactions were performed using 2,2′-azino-bis(3-ethylbenzothiazoline-6-sulphonic acid) substrate (ABTS, Sigma-Aldrich, Schnelldorf, Germany) for 30 min in the dark. Absorbances were detected using a Multiskan FC plate reader (Thermo Fischer, Darmstadt, Germany) at 405 nm. Concentrations of TNFα and IL6 were calculated from standard curves using GraphPad 8 (GraphPad Software Inc., Boston, MA, USA). 

### 2.9. Statistics

All data are given as means ± standard error of the mean (SEM). Multiple-group analysis was performed using one-way ANOVA followed by Tukey´s multiple comparisons test. All statistical analyses were performed using GraphPad Prism 8 (GraphPad Software Inc., Boston, MA, USA) and *p*-values < 0.05 were considered to be statistically significant. 

## 3. Results

### 3.1. Treatment with 2,4D Reduces Number of TH^+^ Dopaminergic Neurons in E14 Midbrain Neuron-Enriched Cultures

In order to evaluate whether 2,4D is directly able to induce the degeneration of mDA neurons, E14 neuron-enriched cultures were treated with 1 mM and 10 µM 2,4D solved in ethanol (EtOH). The concentrations were chosen based on a previous study addressing 2,4D neurotoxicity in cerebellar granule cells [12]. EtOH alone was used to exclude the toxic effects of the solvent. Figure 1A shows the workflow of the experimental design. After 48 h of treatment, the cells were fixed and the mDA neurons were visualized using TH immunocytochemistry. Representative images showing the TH^+^ mDA neurons from the neuron-enriched cultures after different treatments are depicted in Figure 1B–E. The treatment with 2,4D at a concentration of 1 mM resulted in a reduced number of mDA neurons and shortened processes (Figure 1D). 

The quantification of the TH^+^ neurons is given in Figure 1F. Whereas EtOH alone (92.06% ± 1.765%) had no significant effect on neuron survival, 1 mM 2,4D significantly decreased the number of mDA neurons after 48 h (72.02% ± 1.8%). In addition, 10 µM 2,4D only resulted in a slight decrease in TH^+^ neurons (86.77% ± 6.142%), without reaching significance. We further analyzed the levels of TNFα in the supernatants of the treated E14 neuron-enriched cultures. As shown in Figure 1G, treatment with EtOH or 2,4D did not result in a significant release of TNFα compared to the control cells. Notably, the levels of TNFα in the neuron-enriched cultures were close to the detection limit of the ELISA kit, indicating that the TNFα levels in this culture setup are extremely low. These data indicate that 2,4D has direct neurotoxic effects on TH^+^ mDA neurons.

### 3.2. Increased Neurotoxicity of 2,4D in E14 Midbrain Neuron-Glia Cultures

We have recently shown that reactive microglia are able to induce the degeneration of mDA neurons in E14 neuron-glia cultures [19,23]. In order to analyze whether the 2,4D-induced degeneration of mDA neurons results in microglia reactivity and, thus, an enhanced loss of TH^+^ neurons, neuron-glia cultures were prepared and treated in the same way as the neuron-enriched cultures (Figure 2A). Interestingly, we observed that the treatment with 1 mM 2,4D resulted in a more severe loss of TH^+^ neurons, with only a small fraction of surviving cells after 48 h (19.93% ± 5.61%). Again, EtOH alone and 10 µM 2,4D had no significant effect on the numbers of mDA neurons, as indicated by the 87.1% (± 3.685%) and 93.16% (±1.228%) surviving neurons compared to the control cultures (Figure 2B–F). As microglia reactivity is linked to the secretion of inflammatory factors, we further analyzed the level of TNFα in the supernatants of the treated E14 neuron-glia cultures. As shown in Figure 2G, the treatment with 1 mM 2,4D resulted in a significant release of TNFα (0.2385 ng ± 0.01549 ng) compared to the control, EtOH and 10 µM 2,4D treatments. Taken together, these data demonstrate that the 2,4D-induced loss of TH^+^ mDA neurons is enhanced in the presence of microglia in the culture setup. Moreover, 2,4D-induced neuron loss might trigger microglia reactivity and the release of microglia TNFα, which in turn might at least partially contribute to the enhanced neuron loss in mixed neuron-glia cultures. 

### 3.3. Microglia Reactivity Is Not Directly Triggered by 2,4D Application

To exclude the possibility that the microglia reactivity observed in the neuron-glia cultures was not mediated by 2,4D itself, primary microglia cultures were treated with 2,4D or Interferon-γ (IFNγ) and the microglia proliferation and TNFα, as well as IL6 secretion, were analyzed (Figure 3A). As a first approach, the primary microglia were treated with IFNγ (10 ng/mL) or 2,4D (1 mM) for 24 h and the cells were then fixed and labelled using FITC-coupled isolectin. As shown in Figure 3B, the treatment with IFNγ (10 ng/mL) resulted in a more round-shaped microglia morphology, indicative of microglia activation, which was not observed after the treatment with 2,4D. However, changes in the microglia morphology do not precisely predict microglial reactive states in vitro or in vivo. Further, microglia reactivity has been shown to result in the increased proliferation and release of inflammatory factors such as TNFα and IL6 [19,23,24,25]. Therefore, the primary microglia were treated with IFNγ (1 ng/mL and 10 ng/mL) and the cell viability, as an indirect assay for monitoring proliferation, as well as the release of TNFα and IL6 were analyzed in order to validate the successful activation and reactivity of microglia (Figure 3C–E). Treatment with IFNγ (10 ng/mL) for 24 h significantly increased the numbers of viable microglia (168.5 % ± 8.709%), as well as the secretion of TNFα (146.6% ± 16.35%) and IL6 (131.5% ± 11.33%). Next, the primary microglia cultures were left untreated or were incubated with EtOH (as solvent control) and 2,4D (1 mM and 10 µM) for 24 h under serum-free conditions. The viability of the microglia was detected using the MTT assay, and TNFα and IL6 secretion were assessed using TNFα and IL6 ELISA Kits. Figure 3F demonstrates that the treatment with 2,4D was neither able to increase the numbers of viable microglia nor able to stimulate the release of TNFα (Figure 3G) or IL6 (Figure 3H). In summary, these data clearly demonstrate that 2,4D has no direct effect on primary microglia reactivity. The observed increases in TNFα secretion in the neuron-glia cultures was thus likely due to the microglia reactivity triggered by the 2,4D-induced degeneration of mDA neurons.

## 4. Discussion

In the present study, we provide evidence that 2,4D is a neurotoxic agent for TH^+^ mDA neurons in vitro. Moreover, we demonstrate that the 2,4D-induced degeneration of dopaminergic neurons seems to triggers microglia reactivity, as evidenced by the release of TNFα in the neuron-glia cultures. This microglia reactivity might explain the observation of the enhanced loss of mDA neurons in the neuron-glia cultures. We finally reveal that 2,4D alone is not able to directly trigger microglia activation or TNFα and IL6 release in primary microglia cultures, further supporting the notion that neuron damage-induced microglia reactivity is as least partially responsible for the increased 2,4D-driven degeneration of midbrain dopaminergic neurons. 

The risk of developing PD is significantly increased in occupations that use pesticides and herbicides [8]. 2,4D and its derivates are herbicides used to limit the growth of broad leaf weeds all over the world. Exposure to 2,4D has been reported to induce an adverse range of health side effects, including embryotoxicity, teratogenic effects and neurotoxicity [9,10,11]. In vitro studies have demonstrated that 2,4D is able to induce the programmed cell death of cerebellar granule cells [12,13]. Moreover, disruption of the cytoskeleton and disorganization of the Golgi apparatus seem to be the underlying molecular mechanisms of 2,4D-mediated neurotoxicity [14] in this cerebellar neuron population. The knowledge regarding the extent to which dopaminergic neurons and/or the nigrostriatal system itself are prone to 2,4D-induced neurodegeneration is limited. Intraperitoneal injections of 2,4D in adult mice and rabbits have demonstrated the accumulation and regional distribution of 2,4D within the brain without a generalized increase in blood-brain barrier permeability [26]. Moreover, the acute and oral administration of 2,4D in rats was able to decrease locomotion and rearing frequencies and to increase immobility duration observed in an open-field test [16]. The impact of 2,4D administration on the monoamine system has been addressed in previous in vivo studies. The effects of biogenic amine levels in adult rats were analyzed after pre- and post-natal, as well as acute, exposure of adult rats to 2,4D. Both studies detected significant changes in the 5-HT (5-hydroxytryptamine) and DA (dopamine) levels, which explains the above-mentioned motor phenotypes [17,27]. However, the degeneration of mDA neurons has not been reported after this application of 2,4D in vivo. Interestingly, intracerebral injections of 2,4D into rat brains have been demonstrated to result in region-specific neurotoxicity in the nigrostriatal system [15], indicating that 2,4D has a neurotoxic effect in vivo. The mDA neuron cultures used in the present study are known to be very sensitive to neurotoxic stimuli, and have thus been widely used to study neurotoxic substances and/or neurotrophic factors [18,19,23,28,29]. It is well-accepted that these neuronal cultures are stressed in vitro, and therefore might be more prone to degeneration than mDA neurons in vivo. Another factor, especially when considering PD development, is aging. The degeneration of mDA neurons in PD is age-dependent [1], and it is likely that aged mDA neurons might react differently to 2,4D. Moreover, the region-specificity and aging of microglia, particularly in the nigrostriatal system, might also contribute to the increased susceptibility of mDA neurons to degenerate upon contact to neurotoxic agents such as 2,4D [30,31,32].

In the present study, we clearly demonstrate that 2,4D is neurotoxic to midbrain mDA neurons in vitro. Using neuron-enriched cultures, we reveal that 2,4D directly induces neurotoxicity as glial cell populations are missing in this cell culture setup. However, in a mixed neuron-glia environment, the 2,4D-induced neurotoxic effects were enhanced. Recent studies have proven that the involvement of neuroinflammatory responses can increase the loss of mDA neurons in PD patients, animal models of the disease and in vitro models [7,19,33,34,35]. It has been demonstrated that mDA neurons are more susceptible to neurodegeneration induced by neurotoxins such as rotenone, MPTP and paraquat in the presence of microglia [6,36,37]. Microglia react to neurodegeneration and start to secrete factors such as TNFα, IL6, IL1β and nitric monoxide (NO), which in turn are suspected to further mediate the neurotoxic effects [36,37,38,39].

In the present study, we showed that the 2,4D treatment of E14 midbrain neuron-glia cultures resulted in enhanced mDA neurotoxicity, associated with an increased level of secreted TNFα. However, the extent to which other microglia-released factors, such as IL6, IL1β and nitric monoxide (NO), contribute to the observed effect in neuron-glia cultures remains to be elucidates. A direct contribution of IL1β1 is unlikely as the activation of the NLPR3 inflammasome is necessary to process and release mature IL1β1. Further studies are needed to identify the microglia-secreted factors and molecules that enhance the degeneration of mDA neurons in vitro and in vivo. Interestingly, we have recently reported that the activation of microglia with IFNγ and the release of TNFα in this cell culture setup resulted in a substantial loss of mDA neurons. The treatment of neuron-enriched cultures, however, did not result in reduced mDA neuron survival [19]. It is likely that the observed increase in 2,4D-induced neurotoxicity in the neuron-glia cultures in the present study is mediated by microglia reactivity. As 2,4D alone was not able to trigger primary microglia reactivity, we assume that the degeneration of mDA neurons subsequently results in the activation of microglia in neuron-glia cultures. However, while primary microglia reactivity was not triggered by the direct application of 2,4D, a recent study has demonstrated that the microglia cell line BV2 responded to 2,4D through the expression and release of inflammatory factors [40]. These different observations might be explained by the distinct activation states of both cell types. It has recently been shown that BV2 cells display a pre-activated phenotype when compared with primary microglia or ex vivo isolated mouse microglia [41]. 

TNFα could be a very promising molecule fostering further degeneration of mDA neurons. It has been demonstrated that the LPS-induced secretion of TNFα increases activity-dependent neuronal apoptosis in the postnatal cerebral cortex [24]. Moreover, data from mutant mice indicate that TNFα signaling might contribute to the degeneration of mDA neurons in vivo. The knockout of *Tnfα* protected mice from MPTP-induced mDA neuron loss and further resulted in a reduction in neuroinflammatory responses [42]. Sriram and colleagues have demonstrated that mice deficient in *Tnfαr1* and *Tnfαr2* are protected against MPTP-induced neurotoxicity [43,44]. Moreover, the contribution of TNFa to the degeneration of mDA neurons in PD models has been demonstrated in vivo and in vitro. The secretion of TNFa has been detected after the application of the neurotoxin 6-OHDA (6-hydroxydopamine) in mDA neuron cultures. The blocking of TNFa signaling using a pharmacological inhibitor abrogated the loss of mDA neurons [45]. Microglia are supposed to be the source of TNFa in this scenario, as well as in our recent study. However, astrocytes are also known to release TNFa after stimulation with inflammatory factors such as LPS of IFNγ [46]; thus, GFAP^+^ astrocytes could also contribute to the increased levels of TNFa in mDA neuron-glia cultures. The cell–cell-communication within the analyzed cell culture setups used in this study are not well understood, but might be important for the observed effects. At the least, the presence of microglia is necessary to drive neurodegeneration in mDA neuron cultures as microglia-conditioned medium alone is not able to induce the degeneration of mDA neurons [23]. However, the observed increase in the TNFα levels after the treatment of neuron-glia cultures with 2,4D indicate that TNFα could be a potential player triggering the increased loss of mDA neurons. Indeed, it remains unclear how the degeneration of mDA neurons triggers microglia reactivity in the current neuron-glia culture setting. However, the setup of this preliminary study has limitations, such as the different ages of both the mDA neuron culture conditions, different microglia/glia reactive states within these cultures, limited availability of primary mDA neurons to conduct detailed experiments on dose-dependent 2,4D effects and, finally, the in vitro situation itself, which cannot resemble the complex in vivo conditions. 

Nevertheless, the data presented in this study provide evidence that 2,4D has direct neurotoxic effects on mDA neurons in vitro. To the best of our knowledge, this is the first report describing the degeneration of mDA neurons upon 2,4D treatment. Moreover, we demonstrate that 2,4D-driven neurotoxicity triggers the secretion of TNFα, which could be due to microglia reactivity and, thereby, enhances 2,4D-driven neurotoxicity in neuron-glia cultures. In summary, the findings of this preliminary report introduce 2,4D as a direct neurotoxic agent for mDA neurons and further support the hypothesis that microglia substantially contribute to neuroinflammation-mediated neurodegeneration in PD.

## Figures and Tables

**Figure 1 biomedicines-11-02882-f001:**
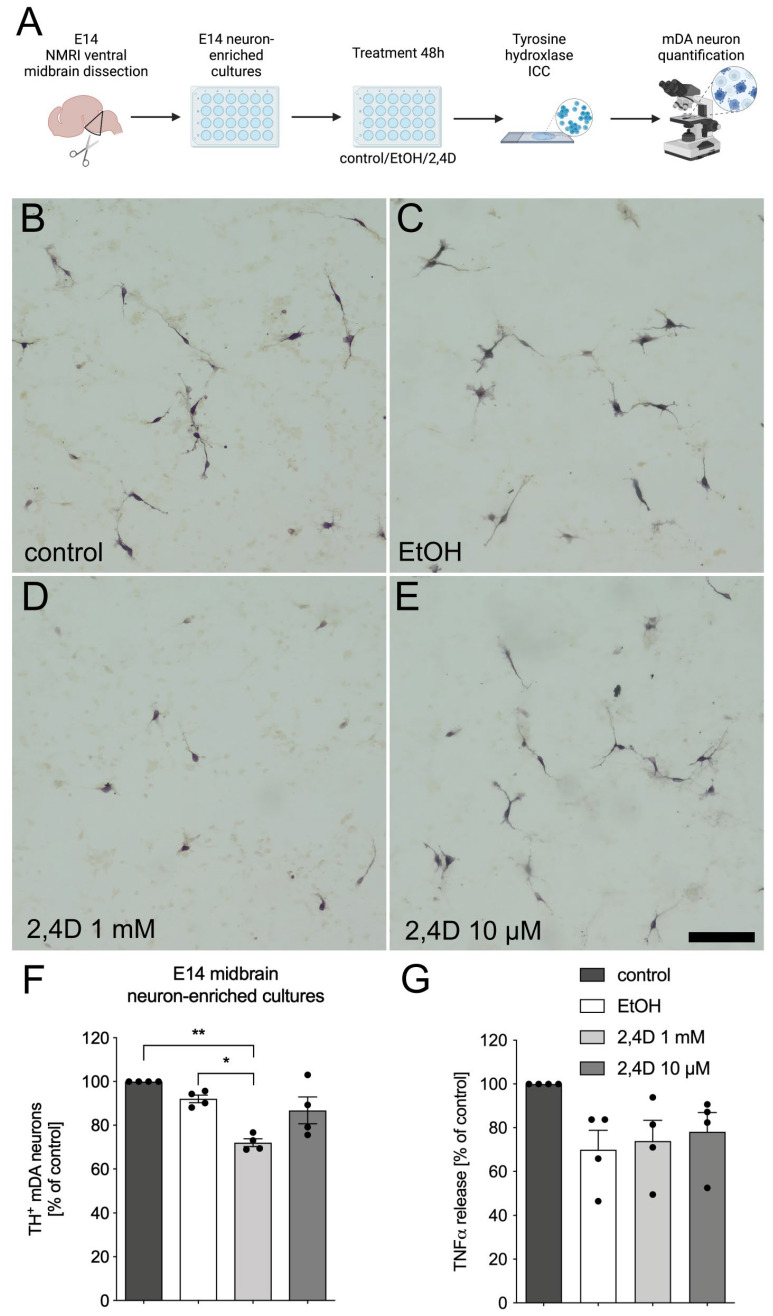
2,4D treatment results in reduced number of TH^+^ neurons in E14 midbrain neuron-enriched cultures. Scheme depicting experimental workflow (**A**). Created using BioRender.com. Neuron-enriched cultures were left untreated as control cells (**B**) or were treated with ethanol (EtOH, (**C**)), 2,4D 1 mM (**D**) and 2,4D 10 µM (**E**) for 48 h. Representative images of TH^+^ neurons for each treatment group are depicted. Scale bar indicates 100 µm. Quantification of TH^+^ neurons in different treatment groups (**F**). Levels of TNFα in supernatants from neuron-enriched cultures after indicated treatments (**G**). Data are given as means ± SEM from at least three independent experiments. *p*-values derived from one-way ANOVA followed by Tukey´s multiple comparisons tests are * *p* < 0.05 and ** *p* < 0.01.

**Figure 2 biomedicines-11-02882-f002:**
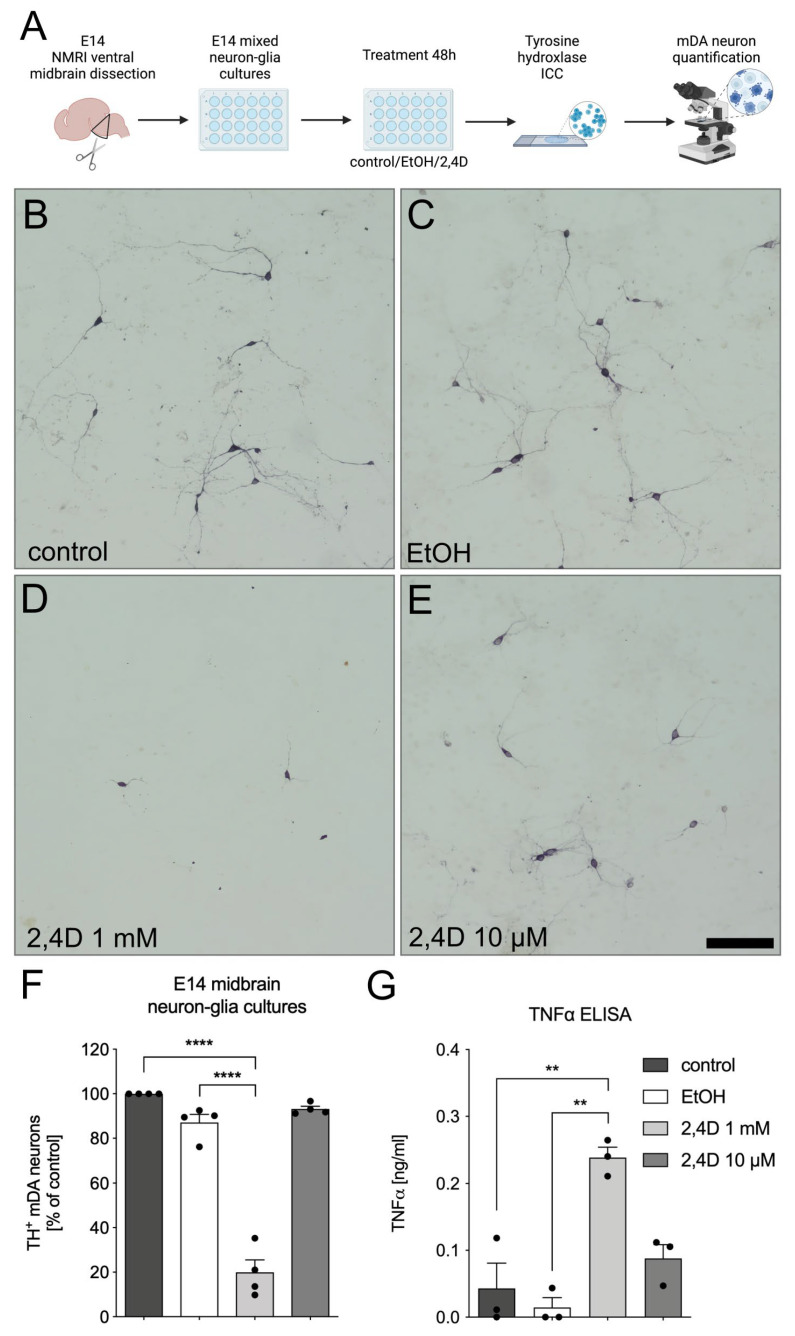
Enhanced loss of TH^+^ neurons in E14 midbrain neuron-glia cultures after 2,4D treatment. Scheme depicting experimental workflow (**A**). Created using BioRender.com. Neuron-glia cultures were left untreated (**B**) or were treated with ethanol (EtOH, (**C**)), 2,4D 1mM (**D**) and 2,4D 10 µM (**E**) for 48 h under serum-free conditions. Representative images of labelled TH^+^ neurons for each treatment group are shown. Scale bar indicates 100 µm. Quantification of TH^+^ neurons in different treatment groups (**F**). TNFα levels in supernatants after indicated treatments (**G**). Data are given as means ± SEM from at least three independent experiments. *p*-values derived from one-way ANOVA followed by Tukey´s multiple comparisons tests are ** *p* < 0.01 and **** *p* < 0.0001.

**Figure 3 biomedicines-11-02882-f003:**
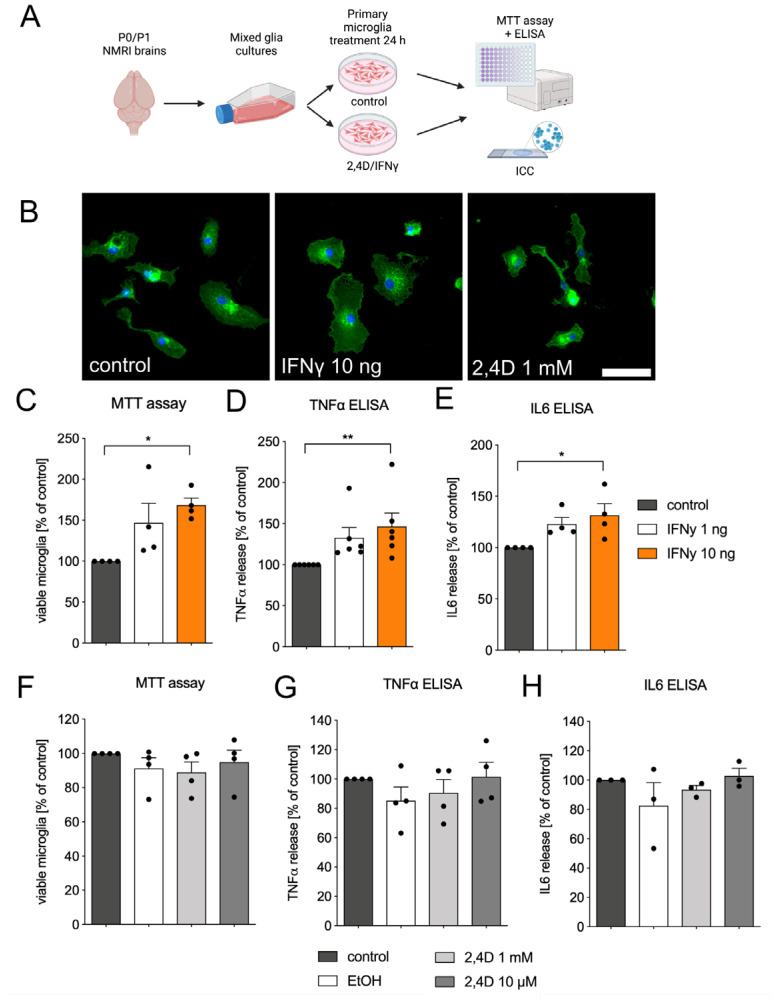
2,4D treatment of microglia cultures is not sufficient to trigger primary microglia reactivity. Scheme showing experimental workflow for primary microglia cultures (**A**). Created using BioRender.com. Microglia cultures were left untreated or were treated with IFNγ 10 ng/mL or 2,4D 1 mM, and microglia morphology after IB4-labelling (green) is depicted as a proxy for microglia reactivity (**B**). Scale bar indicates 50 µm. Treatment of microglia with IFNγ at 1 ng/mL or 10 ng/mL for 24 h under serum-free conditions resulted in increased microglia numbers (**C**) and increased release of TNFα (**D**) and IL6 (**E**). No changes in numbers of viable microglia (**F**) and microglial TNFα (**G**) and IL6 (**H**) secretion could be observed after treatment with 2,4D for 24 h. Data are given as means ± SEM from at least three independent experiments. *p*-values derived from one-way ANOVA followed by Tukey´s multiple comparisons tests are * *p* < 0.05 and ** *p* < 0.01.

## Data Availability

The data that support the findings of this study are available from the corresponding author upon reasonable request.

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
