# Peer review of "2,4-Dichlorophenoxyacetic Acid Induces Degeneration of mDA Neurons In Vitro"

_biomedicines, 2023, doi:10.3390/biomedicines11112882_

Round 1
Reviewer 1 Report
Studying the mechanisms of development of Parkinson's disease is undoubtedly an urgent task in neurobiology. The authors described The neurotoxic features of 2,4-Diclorophenoxyacetic acid in an in vitro model. The article is interesting, relevant and, in my opinion, can be accepted for publication in the journal Biomedicines. However, there are a few points that need to be clarified.
1. It is necessary to indicate what the final concentration of ethanol was in 10 µM and 1 mM 2,4-Diclorophenoxyacetic. Since, in general, the ethanol concentration when treating cells should not exceed 0.1%.
2. Indicate what concentration of ethanol was used EtOH?
3. In the methods of statistical analysis, it is necessary to indicate which test was used when conducting One-Way Anova.
4. Why only TNF alfa was used to study the proinflammatory activity of the test compound, it seems to me that it would be more appropriate to look at a wider range of proinflammatory cytokines (for example, IL1, IL6, INF gamma).
5. Line 215, it seems the parentheses with numbers are placed incorrectly.
Author Response
Studying the mechanisms of development of Parkinson's disease is undoubtedly an urgent task in neurobiology. The authors described The neurotoxic features of 2,4-Diclorophenoxyacetic acid in an in vitro model. The article is interesting, relevant and, in my opinion, can be accepted for publication in the journal Biomedicines. However, there are a few points that need to be clarified.
1. It is necessary to indicate what the final concentration of ethanol was in 10 µM and 1 mM 2,4-Diclorophenoxyacetic. Since, in general, the ethanol concentration when treating cells should not exceed 0.1%.
Response: Thanks for this valuable comment. When using 2,4D concentrations of 1 mM the concentration of ethanol in the culture medium was 0.043%. When using 10 µM the concentration was 0.00043%. We have added this important information to the methods section.
2. Indicate what concentration of ethanol was used EtOH?
Response: 2,4D solutions were prepared using 95% Ethanol. 100mM working solutions have been used. We added this information to the methods section of the manuscript.
3. In the methods of statistical analysis, it is necessary to indicate which test was used when conducting One-Way ANOVA.
Response: Tukey´s multiple comparisons test was used in combination with one-way ANOVA. We have added this information to the methods section.
4. Why only TNF alfa was used to study the proinflammatory activity of the test compound, it seems to me that it would be more appropriate to look at a wider range of proinflammatory cytokines (for example, IL1, IL6, INF gamma).
Response: Thank you for this comment. We initially focused on TNFa since it has been linked to mDA neuron death in PD models [1–3]. However, we have added IL6 ELISA data to Figure 3 in order to better validate to proinflammatory activity of microglia.
5. Line 215, it seems the parentheses with numbers are placed incorrectly.
Response: Thanks for this comment. We have checked the parentheses with numbers and found them to be correct.
1. Block, M.L.; Zecca, L.; Hong, J.-S. Microglia-Mediated Neurotoxicity: Uncovering the Molecular Mechanisms. Nat. Rev. Neurosci. 2007, 8, 57–69, doi:10.1038/nrn2038.
2. Qin, L.; Wu, X.; Block, M.L.; Liu, Y.; Breese, G.R.; Hong, J.-S.; Knapp, D.J.; Crews, F.T. Systemic LPS Causes Chronic Neuroinflammation and Progressive Neurodegeneration. Glia 2007, 55, 453–462, doi:10.1002/glia.20467.
3. Qin, L.; Liu, Y.; Wang, T.; Wei, S.-J.; Block, M.L.; Wilson, B.; Liu, B.; Hong, J.-S. NADPH Oxidase Mediates Lipopolysaccharide-Induced Neurotoxicity and Proinflammatory Gene Expression in Activated Microglia. J. Biol. Chem. 2004, 279, 1415–1421, doi:10.1074/jbc.M307657200.
Reviewer 2 Report
This manuscript by Tamara Russ and colleagues reports on the neurotoxic effect of 2,4-Diclorophenoxyacetic acid (2,4-D) by inducing degeneration of the primary dopaminergic neuron cells in the midbrain. In brief, 2,4-D induced neurodegeneration by promoting the TNF alpha release from microglia through unknown mechanism in the primary culture system.
Although the study provides an important contribution to elucidation of a mechanism of herbicide-induced neurodegeneration in PD, the purpose of this study has not been fully archived as presented. The additional experiments are needed for the publication.
Major issues:
1. The authors indicated that 2,4-D induced a release of TNF alpha by activating microglia, resulting in enhanced neuron loss in the neuron-glia cell culture. This result is a critical point in this article, so that the authors need to clarify the involvement of TNF alpha in this assay system. For example, it can be verified by adding recombinant TNF alpha to the neuron-enriched cultures, and see whether neuron loss is enhanced.
2. The authors insisted that TNF alpha release from microglia by 2,4-D-induced neuron loss caused further neuronal damage. Again, the involvement of TNF alpha on neurotoxic effect should be shown more directly. For instance, primary glia cells from TNF-knockout mice or TNF knocked down cells could be useful tools for the verification of the hypothesis.
3. The author labelled primary microglia with isolectin antibodies and identified microglial activation by observing its morphology. Normally, the microglial markers such as Iba-1 should be used for the detection of microglial activation.
4. The author showed only one-way ANOVA results for the statistical analysis. However, the post-hoc test is also needed for the multiple comparison and addressed somewhere in methods section or figure legends.
Minor issues:
1. Is there any reason for addressing “NMRI” mice in figures and result section? Is it just a wild-type mouse or a specific mouse for this article?
2. The authors treated primary cultures with 1mM or 10uM concentration of 2,4-D. The author had better explain the reason for selecting these concentrations. The referenced paper performed the experiment using 1mM or 2mM concentration but not 10uM. Otherwise, it would be interesting to show the dose-dependency of 2,4-D treatment, e.g., 10uM, 100uM, 1mM and 2mM.
3. The methods section and figure legends should be described precisely. For instance, the number of cells at the starting point, relevant statistical analysis, the description of mice, Animal ethics and so on.
Author Response
This manuscript by Tamara Russ and colleagues reports on the neurotoxic effect of 2,4-Diclorophenoxyacetic acid (2,4-D) by inducing degeneration of the primary dopaminergic neuron cells in the midbrain. In brief, 2,4-D induced neurodegeneration by promoting the TNF alpha release from microglia through unknown mechanism in the primary culture system.
Although the study provides an important contribution to elucidation of a mechanism of herbicide-induced neurodegeneration in PD, the purpose of this study has not been fully archived as presented. The additional experiments are needed for the publication.
Major issues:
- The authors indicated that 2,4-D induced a release of TNF alpha by activating microglia, resulting in enhanced neuron loss in the neuron-glia cell culture. This result is a critical point in this article, so that the authors need to clarify the involvement of TNF alpha in this assay system. For example, it can be verified by adding recombinant TNF alpha to the neuron-enriched cultures, and see whether neuron loss is enhanced.
Response: The authors are grateful for this comment and totally agree that the involvement of TNFa was not precisely addressed in this manuscript. Several previous studies have shown that TNFa is essential to drive LPS-induced degeneration of mDA neurons in vivo [1,2] and in vitro [3]. We have further shown that LPS treatment as well as IFNgonly results in loss of mDA neurons in neuron-glia cultures whereas neuron-enriched were not affected by LPS or IFNgtreatment [4,5]. The same culture models have been used in the current manuscript. Based on these studies it is high likely that TNFa could also be a key player in mDA degeneration in the present study. We have further added ELISA data to Figure 1 showing TNFa levels in neuron-enriched cultures. TNFa was hardly detectable in this culture setup and no significant changes were observed after treatment with 2,4D. Unfortunately, due to the short time (10 days) given for the revision of the manuscript, no additional experiments were possible. Thus, we have reduced the emphasis on TNFa throughout the manuscript rather suggesting its involvement.
- The authors insisted that TNF alpha release from microglia by 2,4-D-induced neuron loss caused further neuronal damage. Again, the involvement of TNF alpha on neurotoxic effect should be shown more directly. For instance, primary glia cells from TNF-knockout mice or TNF knocked down cells could be useful tools for the verification of the hypothesis.
Response: The authors completely agree to this reviewer´s comment and again, we were not able to include additional experiments as suggested by the reviewer´s due to the short revision deadline. However, we agree that future studies should elucidate the contribution of TNFa to mDA neurodegeneration. Again, we have rephrased the manuscript in the way that our data suggest that TNFa could be an important factor explaining increased neuron loss in neuron-glia cultures.
- The author labelled primary microglia with isolectin antibodies and identified microglial activation by observing its morphology. Normally, the microglial markers such as Iba-1 should be used for the detection of microglial activation.
Response: Indeed, Iba1 is a commonly used marker for microglia. However, antibody-free labelling with isolectins has been proven to be a reliable and specific method to label microglia in vitro as well in vivo [6–8]
- The author showed only one-way ANOVA results for the statistical analysis. However, the post-hoc test is also needed for the multiple comparison and addressed somewhere in methods section or figure legends.
Response: Thank you for this valuable comment. Tukey´s multiple comparisons test was used in combination with one-way ANOVA. We have added this information to the methods section.
Minor issues:
- Is there any reason for addressing “NMRI” mice in figures and result section? Is it just a wild-type mouse or a specific mouse for this article?
Response: When neuron cultures are prepared from E14 mouse embryos, C57BL/6 or NMRI outbred strains are commonly used. Since both strains have also been shown to react differently in several studies, we added the detailed information throughout the manuscript in order to make clear what exact mouse strain was used for each experiment.
- The authors treated primary cultures with 1mM or 10uM concentration of 2,4-D. The author had better explain the reason for selecting these concentrations. The referenced paper performed the experiment using 1mM or 2mM concentration but not 10uM. Otherwise, it would be interesting to show the dose-dependency of 2,4-D treatment, e.g., 10uM, 100uM, 1mM and 2mM.
Response: The authors thank the reviewer for this comment. Indeed, these two different concentrations are within a wide range. We have decided to use a “standard” concentration with 1 mM as previously described. However, since mDA neurons in culture have been shown to be very sensitive to several toxins, we decided to include a rather low concentration of 2,4D in order to analyze whether mDA neurons might be sensitive to this low concentration. We agree, that it would be interesting to analyze dose-dependent effects, however, the yield of neuronal cultures from E14 ventral midbrains is very low and the amounts of cells for this kind of study would be huge.
- The methods section and figure legends should be described precisely. For instance, the number of cells at the starting point, relevant statistical analysis, the description of mice, Animal ethics and so on.
Response: Thank you very much for this advice. We have added the listed missing information where appropriate and the methods section of the manuscript has been updated accordingly.
1. Block, M.L.; Zecca, L.; Hong, J.-S. Microglia-Mediated Neurotoxicity: Uncovering the Molecular Mechanisms. Nat. Rev. Neurosci. 2007, 8, 57–69, doi:10.1038/nrn2038.
2. Qin, L.; Wu, X.; Block, M.L.; Liu, Y.; Breese, G.R.; Hong, J.-S.; Knapp, D.J.; Crews, F.T. Systemic LPS Causes Chronic Neuroinflammation and Progressive Neurodegeneration. Glia 2007, 55, 453–462, doi:10.1002/glia.20467.
3. Qin, L.; Liu, Y.; Wang, T.; Wei, S.-J.; Block, M.L.; Wilson, B.; Liu, B.; Hong, J.-S. NADPH Oxidase Mediates Lipopolysaccharide-Induced Neurotoxicity and Proinflammatory Gene Expression in Activated Microglia. J. Biol. Chem. 2004, 279, 1415–1421, doi:10.1074/jbc.M307657200.
4. Zhou, X.; Spittau, B. Lipopolysaccharide-Induced Microglia Activation Promotes the Survival of Midbrain Dopaminergic Neurons In Vitro. Neurotox Res 2018, 33, 856–867, doi:10.1007/s12640-017-9842-6.
5. Zhou, X.; Zöller, T.; Krieglstein, K.; Spittau, B. TGFβ1 Inhibits IFNγ-Mediated Microglia Activation and Protects mDA Neurons from IFNγ-Driven Neurotoxicity. J. Neurochem. 2015, 134, 125–134, doi:10.1111/jnc.13111.
6. Streit, W.J.; Kreutzberg, G.W. Lectin Binding by Resting and Reactive Microglia. J Neurocytol 1987, 16, 249–260, doi:10.1007/BF01795308.
7. Villacampa, N.; Almolda, B.; González, B.; Castellano, B. Tomato Lectin Histochemistry for Microglial Visualization. Methods Mol Biol 2013, 1041, 261–279, doi:10.1007/978-1-62703-520-0_23.
8. Brawek, B.; Olmedillas Del Moral, M.; Garaschuk, O. In Vivo Visualization of Microglia Using Tomato Lectin. Methods Mol Biol 2019, 2034, 165–175, doi:10.1007/978-1-4939-9658-2_12.
Reviewer 3 Report
The manuscript (biomedicines-2659344) describes the toxic effects of the herbicide 2,4-Dichlorophenoxyacetic acid (2,4D) in DIV1 E14 ventral midbrain dopaminergic (mDA) neuron-enriched cultures and in DIV8 E14 ventral midbrain dopaminergic neuron-glia cultures. 2,4D at 1 mM, but not at 10μM was found to cause TH-positive dopamine neuronal loss, which was exacerbated in the neuron-glia culture where increased TNFα levels in supernatants were observed. However, unlike IFNγ, 2,4D did not activate primary microglia culture as shown by morphology and TNFα secretion. It was concluded that 2,4D is neurotoxic for midbrain dopamine neurons, which is enhanced by microglia, and could be a risk factor for Parkinson’s disease (PD).
Overall, while the reviewer finds the involvement of glia in 2,4D effects interesting, the study is very much preliminary and there are significant flaws/deficits in the study design and ignore of previous literature on possible effects of 2,4D on dopamine neurons in vivo. Below are my comments for the authors to address:
1. 2,4D was used at 10 µM and 1 mM, with the latter found to be effective. However, the mM range appears unlikely to be achievable in vivo in brain based on the reports of Kim et al (Brain Research, 1988, 440:216-226) and Oliveira and Palermo-Neto (Pharmacology & Toxicology 1993, 73, 19-85). Moreover, chronic 2,4D at 70 mg/kg (oral) did not result in significantly striatal dopamine loss in brain (Bortolozzi A, Duffard R, de Duffard AM. Asymmetrical Development of the Monoamine Systems in 2,4-Dichlorophenoxyacetic Acid Treated Rats. Neurotoxicology. 2003 Jan;24(1):149-57), which casts doubts that at a more achievable level 2,4D is neurotoxic to dopamine neurons in vivo or could have caused PD. The concentration used should be included in the abstract and previous literature on 2,4D in animals in vivo, in particular on dopamine, should be cited and have a balanced discussion.
2. Could older culture (DIV 1 vs DIV 8) be a factor in the difference of the two experiments? This should have been matched in the design.
3. What was the neuronal morphology after 2,4D? And how about microglia and astrocytes in the cultures? Why was TNFα not measured in experiment #1? What are the sources of TNFα? Is it possible by astrocytes too?
4. Why was 24 hr used for the treatment of primary microglia culture, in contrast to 48 hr for the neurons/neuron-glia cultures? A recent study (Zhou et al. 2022 Environ Toxicol 2022 May;37(5):1136-1151) reported activation of BV2 microglia by 2,4D at a similar concentration. Although primary microglia might respond differently, the current manuscript is limited by examination of only TNFα.
5. What are the detection limit and coefficient of variance for the TNFα assay? What are the control levels of TNFα in experiment 3?
6. Could 2,4D have affected the growth and maturation of the mDA neurons instead of killing them?
7. Mis-spelling ‘2,4-Diclorophenoxyacetic acid’ in many cases including the title.
Author Response
The manuscript (biomedicines-2659344) describes the toxic effects of the herbicide 2,4-Dichlorophenoxyacetic acid (2,4D) in DIV1 E14 ventral midbrain dopaminergic (mDA) neuron-enriched cultures and in DIV8 E14 ventral midbrain dopaminergic neuron-glia cultures. 2,4D at 1 mM, but not at 10μM was found to cause TH-positive dopamine neuronal loss, which was exacerbated in the neuron-glia culture where increased TNFα levels in supernatants were observed. However, unlike IFNγ, 2,4D did not activate primary microglia culture as shown by morphology and TNFα secretion. It was concluded that 2,4D is neurotoxic for midbrain dopamine neurons, which is enhanced by microglia, and could be a risk factor for Parkinson’s disease (PD). Overall, while the reviewer finds the involvement of glia in 2,4D effects interesting, the study is very much preliminary and there are significant flaws/deficits in the study design and ignore of previous literature on possible effects of 2,4D on dopamine neurons in vivo. Below are my comments for the authors to address:
- 2,4D was used at 10 µM and 1 mM, with the latter found to be effective. However, the mM range appears unlikely to be achievable in vivo in brain based on the reports of Kim et al (Brain Research, 1988, 440:216-226) and Oliveira and Palermo-Neto (Pharmacology & Toxicology 1993, 73, 19-85). Moreover, chronic 2,4D at 70 mg/kg (oral) did not result in significantly striatal dopamine loss in brain (Bortolozzi A, Duffard R, de Duffard AM. Asymmetrical Development of the Monoamine Systems in 2,4-Dichlorophenoxyacetic Acid Treated Rats. Neurotoxicology. 2003 Jan;24(1):149-57), which casts doubts that at a more achievable level 2,4D is neurotoxic to dopamine neurons in vivo or could have caused PD. The concentration used should be included in the abstract and previous literature on 2,4D in animals in vivo, in particular on dopamine, should be cited and have a balanced discussion.
Response: The authors thank the reviewer for this important point. We have included these essential missing references into the discussion to keep it more balanced. The concentrations of 2,4D used throughout the manuscript haven been included in the abstract.
Could older culture (DIV 1 vs DIV 8) be a factor in the difference of the two experiments? This should have been matched in the design.
Response: The authors thank the reviewer for this excellent and valuable comment. Indeed, the different age of the neurons in these two different culture setups should be taken into account. We have added a short part for this in the discussion part of the manuscript. However, at this point we are not able to exclude that this might be a factor. Recent studies from our group using these culture models have proven their potential in addressing the role of glial contribution to mDA neurodegeneration in vitro [4,5,9,10]. Therefore, we are convinced that the different age of mDA neurons in this experimental will only marginally contribute to the observed effects.
- What was the neuronal morphology after 2,4D? And how about microglia and astrocytes in the cultures? Why was TNFα not measured in experiment #1? What are the sources of TNFα? Is it possible by astrocytes too?
Response: The neuronal morphology was not in detail analyzed in the present study. However, there was a clear change in neuron morphology after 1 mM 2,4D treatment. Shorter processes and less ramification could be observed as a proxy for neuronal stress and damage. We have addressed the cellular composition of both neuron culture models in a recent study [11] showing very low numbers of GFAP+ astrocytes and Iba1+ microglia in neuron-enriched cultures but high number of astrocytes and microglia in neuron-glia cultures.
We have further added ELISA data to Figure 1 showing TNFa levels in neuron-enriched cultures. TNFa was hardly detectable in this culture setup and no significant changes were observed after treatment with 2,4D. Regarding the sources of TNFa, we cannot exclude that astroglia might contribute to the observed increased levels of TNFa. We have addressed this issue also in the revised discussion. Again, thank you very much for this comment.
- Why was 24 hr used for the treatment of primary microglia culture, in contrast to 48 hr for the neurons/neuron-glia cultures? A recent study (Zhou et al. 2022 Environ Toxicol 2022 May;37(5):1136-1151) reported activation of BV2 microglia by 2,4D at a similar concentration. Although primary microglia might respond differently, the current manuscript is limited by examination of only TNFα.
Response: This is indeed a very good point. In several studies using primary mouse microglia we have tried to increase the treatment time longer than 24 h. However, under serum-free conditions we know that microglia start dying after 24h. Therefore, we commonly treat for 24 h. The neuron cultures are different. Neuron-enriched cultures are supplemented with N2 to provide essential survival-promoting factors that keep these cultures alive for several days. Neuron-glia cultures are also stable for a couple of days under serum-free conditions. The reason here is the neurotrophic support provided by the confluent glia cell layer being established after 7 days in vitro when our treatments begin.
The authors further appreciate the introduction of the recent study showing 2,4D-driven activation BV2 cells. Indeed, these cells react different in our experimental setups. In this report we have not addressed 2,4D effects in BV2 cells since we know that this line displays a strong pre-activated microglia phenotype. We have included this study in the discussion. Moreover, we have added IL6 as a microglia activation marker to Figure 3 in order to not solely focus on TNFa.
What are the detection limit and coefficient of variance for the TNFα assay? What are the control levels of TNFα in experiment 3?
Response: We have used murine ELISA Kits for TNFa and IL6 from Peprotech. The detection limit of these kits are 16 pg/ml for for TNFa and 20 pg/ml for IL6. The average intra-assay coefficient of variance for the for TNFa ELISA was 5.6%. For experiment 3, primary microglia, the control concentrations of TNFa were 1,719 ng/ml ± 0.141 ng/ml. Which is a relatively high concentration but commonly observed in cultured microglia.
- Could 2,4D have affected the growth and maturation of the mDA neurons instead of killing them?
Response: This is a very good question. Indeed, growth and maturation could be affected as well. However, our experience with these two mDA neuron culture setups has taught us that loss TH-immunoreactivity as the indicator for degenerating and dying neurons. The cellular morphology with shortened processes and reduced ramifications is further indicative of mDA degeneration. It is normally a two-step process in these cultures after application of neurotoxic agents: retraction of processes followed by cell death. This is a major difference to in vivo application of e.g. MPTP, which can result in loss of TH expression whereas the mDA neurons are still alive and are able to recover by means of re-expression of TH.
- Mis-spelling ‘2,4-Diclorophenoxyacetic acid’ in many cases including the title.
Response: The authors apologize for this typo. It has now been corrected throughout the manuscript.
4. Zhou, X.; Spittau, B. Lipopolysaccharide-Induced Microglia Activation Promotes the Survival of Midbrain Dopaminergic Neurons In Vitro. Neurotox Res 2018, 33, 856–867, doi:10.1007/s12640-017-9842-6.
5. Zhou, X.; Zöller, T.; Krieglstein, K.; Spittau, B. TGFβ1 Inhibits IFNγ-Mediated Microglia Activation and Protects mDA Neurons from IFNγ-Driven Neurotoxicity. J. Neurochem. 2015, 134, 125–134, doi:10.1111/jnc.13111.
9. Spittau, B.; Zhou, X.; Ming, M.; Krieglstein, K. IL6 Protects MN9D Cells and Midbrain Dopaminergic Neurons from MPP+-Induced Neurodegeneration. Neuromolecular Med. 2012, 14, 317–327, doi:10.1007/s12017-012-8189-7.
10. Hühner, L.; Rilka, J.; Gilsbach, R.; Zhou, X.; Machado, V.; Spittau, B. Interleukin-4 Protects Dopaminergic Neurons In Vitro but Is Dispensable for MPTP-Induced Neurodegeneration In Vivo. Front Mol Neurosci 2017, 10, 62, doi:10.3389/fnmol.2017.00062.
11. Machado, V.; Haas, S.J.-P.; von Bohlen Und Halbach, O.; Wree, A.; Krieglstein, K.; Unsicker, K.; Spittau, B. Growth/Differentiation Factor-15 Deficiency Compromises Dopaminergic Neuron Survival and Microglial Response in the 6-Hydroxydopamine Mouse Model of Parkinson’s Disease. Neurobiol. Dis. 2016, 88, 1–15, doi:10.1016/j.nbd.2015.12.016.
Reviewer 4 Report
Here are some suggestions to ameliorate the manuscript.
1. Changes in cell number and viability are used to demonstrate the toxic effects of 2,4-diclorophenoxyacetic acid. However, it is not clear how cell counting was achieved after immunocytochemistry.
2. In addition (see for example fig.3), it appears that cell number was calculated as the result of the MTT test. However, MTT (viability test) couldn’t coincide with cell proliferation, especially under well-defined toxic conditions. This issue requires attention and a better explanation/demonstration.
3. Lines 231-232: “As shown in Fig. 3B, treatment with IFNg resulted in a round-shaped microglia morphology indicative of microglia activation”. Although microglial activation may be accompanied by morphological changes, this cannot accurately predict the microglial cell activities.
4. In lines 304-305 is stated that: “the observed increase in TNFa levels after treatment of neuron-glia cultures with 2,4D indicate that TNFa might be a key player triggering increased loss of mDA neurons.” This interpretation is rather speculative since is not supported by any specific proof, and many other pathways and neuroinflammatory markers, apart from the TNF-α, such as IL-1β, IL-10, nitric oxide, among others, can be claimed as responsible for toxicity. At least this should be discussed, or demonstrate the causative role of TNF-α release; for example, by inhibiting the specific receptors.
5. In line 62 (and elsewhere) is highlighted that “2,4D alone was not capable of directly triggering microglia reactivity”: the cell-cell intercommunication could be an important element, particularly for specific drug development. Wy the Authors didn’t try any experimental approach to deepen this result, for example testing the P2X7 receptor that has been shown to be involved in microglia activation? At least this aspect should be discussed to provide the readers with further ideas.
Author Response
Here are some suggestions to ameliorate the manuscript.
- Changes in cell number and viability are used to demonstrate the toxic effects of 2,4-diclorophenoxyacetic acid. However, it is not clear how cell counting was achieved after immunocytochemistry.
Response: The authors thank the reviewer for this important point. Requested information has been added to the methods section of the manuscript.
- In addition (see for example fig.3), it appears that cell number was calculated as the result of the MTT test. However, MTT (viability test) couldn’t coincide with cell proliferation, especially under well-defined toxic conditions. This issue requires attention and a better explanation/demonstration.
Response: The authors completely agree to this reviewer´s comment. Indeed, MTT allows only to quantify viable cells and is only an indirect approach to analyze proliferation. In the present study we only used MTT in our microglia culture experiments, where we did not see degeneration of microglia cells under control condition and, thus, numbers of viable cells are high likely due to proliferation. Although we agree, that MTT assay is not a precise method to quantify proliferation, it is indeed widely used to monitor microglia proliferative activity [12,13].
- Lines 231-232: “As shown in Fig. 3B, treatment with IFNg resulted in a round-shaped microglia morphology indicative of microglia activation”. Although microglial activation may be accompanied by morphological changes, this cannot accurately predict the microglial cell activities.
Response: The authors agree. We have rephrased the sentence in order to emphasize that the morphology observed is not an adequate predictor for microglia reactivity.
- In lines 304-305 is stated that: “the observed increase in TNFa levels after treatment of neuron-glia cultures with 2,4D indicate that TNFa might be a key player triggering increased loss of mDA neurons.”This interpretation is rather speculative since is not supported by any specific proof, and many other pathways and neuroinflammatory markers, apart from the TNF-α, such as IL-1β, IL-10, nitric oxide, among others, can be claimed as responsible for toxicity. At least this should be discussed, or demonstrate the causative role of TNF-α release; for example, by inhibiting the specific receptors.
Response: Thank you very much for this important point. We agree, that this line is rather speculative and, thus, have changed the sentence to underline its speculative nature. The discussion part of the manuscript has been updated to put the emphasis on the role of microglia reactivity in general rather than speculating on TNFa-driven functions alone. Moreover, other well-described factors such as IL1ß or NO have been included in the discussion part.
Unfortunately, due to the short time (10 days) given for the revision of the manuscript, no additional experiments were possible. Thus, we have reduced the emphasis on TNFa throughout the manuscript rather suggesting its involvement.
- In line 62 (and elsewhere) is highlighted that “2,4D alone was not capable of directly triggering microglia reactivity”: the cell-cell intercommunication could be an important element, particularly for specific drug development. Why the Authors didn’t try any experimental approach to deepen this result, for example testing the P2X7 receptor that has been shown to be involved in microglia activation? At least this aspect should be discussed to provide the readers with further ideas.
Response: The authors are thankful for comment and agree that the cell-cell-communication in neuron-glia cultures is important but difficult to tackle. We are currently addressing this cellular cross-talk (microglia-astrocytes-neurons) in a distinct study.
We have added a section to the discussion addressing cell-cell communication for 2,4D-driven neurotoxicity in our culture setup. Again, thank you very much for all constructive comments and suggestions.
12. Zhang, L.; Ma, P.; Guan, Q.; Meng, L.; Su, L.; Wang, L.; Yuan, B. Effect of Chemokine CC Ligand 2 (CCL2) on Α‑synuclein‑induced Microglia Proliferation and Neuronal Apoptosis. Molecular Medicine Reports 2018, 18, 4213–4218, doi:10.3892/mmr.2018.9468.
13. Schmidt, C.; Schneble-Löhnert, N.; Lajqi, T.; Wetzker, R.; Müller, J.P.; Bauer, R. PI3Kγ Mediates Microglial Proliferation and Cell Viability via ROS. Cells 2021, 10, 2534, doi:10.3390/cells10102534.
Round 2
Reviewer 2 Report
The manuscript has been much imploved. I think this article is now suitable for publication in its current form.
Author Response
Thank you again.
Reviewer 3 Report
The reviewer appreciate the effort of the authors in revising the manuscript. Nevertheless, the manuscript can be improved in several aspects. First, since there is no direct evidence that microglia are the culprit in increased DA neuronal loss in the neuron-glia mixed culture, the manuscript should tone down, in particular in the Abstract, on the possible role of microglia. Second, the Introduction should also cite negative in vivo reports of 2.4D on dopamine so that the literature is balanced. Finally, the Discussion should include a section on many limitations of the current report, including the dose/concentration employed, differential culture age, lack of details on glia in the mixed culture, potentially different status of primary microglia culture, etc. Overall it needs to be emphasized that this is a preliminary report and definitely not conclusive on the toxicity of 2,4D in dopamine neurons.
Author Response
The authors thank the reviewer again for this constructive criticism. First, we have modified the abstract to ton down the role of microglia. Second, we have added the important references describing the 2,4D effects on dopamine in vivo. Finally, we have added the requested paragraph discussing the limitations of the present and emphasized the preliminary nature of the current study.
Reviewer 4 Report
Thank you for the agreements and compromise adopted, the manuscript is now ready for publication. Congratulation.
Author Response
Thank you again for your valuable comments and suggestions.